# Seasonal Differences in *Cyclospora cayetanensis* Prevalence in Colombian Indigenous People

**DOI:** 10.3390/microorganisms9030627

**Published:** 2021-03-18

**Authors:** Hagen Frickmann, Juliane Alker, Jessica Hansen, Juan Carlos Dib, Andrés Aristizabal, Gustavo Concha, Ulrich Schotte, Simone Kann

**Affiliations:** 1Department of Microbiology and Hospital Hygiene, Bundeswehr Hospital Hamburg, 20359 Hamburg, Germany; frickmann@bnitm.de; 2Institute for Medical Microbiology, Virology and Hygiene, University Medicine Rostock, 18057 Rostock, Germany; 3Central Institute of the Bundeswehr Medical Service Kiel, Department A—Veterinary Medicine, 24119 Kronshagen, Germany; juliane.alker@gmail.com (J.A.); U.Schotte@t-online.de (U.S.); 4Department for Infectious Disease Diagnostics, Bernhard-Nocht Institute for Tropical Medicine, 20359 Hamburg, Germany; Jessica_hansen@gmx.de; 5Tropical Health Foundation, Santa Marta 470003, Colombia; jdibdiaz@uninorte.edu.co (J.C.D.); ajaristizabal@fspt.co (A.A.); 6Organization Wiwa Yugumaiun Bunkauanarrua Tayrona (OWYBT), Department Health Advocacy, Valledupar 200001, Colombia; gustavoconcha16@gmail.com; 7Medical Mission Institute, 97074 Würzburg, Germany

**Keywords:** *Cyclospora cayetanensis*, Colombia, indigenous people, prevalence, seasonality, infection, colonization, real-time PCR

## Abstract

Fecal-orally transmitted cyclosporiasis is frequent in remote resource-limited settings in Central and South America with poor hygiene conditions. In this study, we aimed at assessing seasonal effects on the epidemiology of colonization or infection with *C. cayetanensis* in Colombian indigenous people living under very restricted conditions. In the rainy season between July and November and in the dry season between January and April, stool samples from indigenous people with and without gastrointestinal symptoms were collected and screened for *C. cayetanensis* applying in-house real-time polymerase chain reaction (PCR). In the rainy season and in the dry season, positive PCR results were observed for 11.8% (16/136) and 5.1% (15/292), respectively, with cycle threshold (Ct) values of 30.6 (±3.4) and 34.4 (±1.6), respectively. Despite higher parasite loads in the rainy season, fewer individuals (2/16, 12.5%) reported gastrointestinal symptoms compared to the dry season (6/15, 40%). In conclusion, considerable prevalence of *C. cayetanensis* in Colombian indigenous people persists in the dry season. Low proportions of gastrointestinal symptoms along with higher parasite loads make colonization likely rather than infection.

## 1. Introduction

The indigenous tribe called Wiwa inhabits remote territories of the Sierra Nevada de Santa Marta in the north-east of Colombia. Living conditions are simple, e.g., their houses consist of palm roofs, mud floors and mud walls. Drinking water is obtained from the nearby river and/or from open cisterns, to which animals have access as well. Sanitation, electricity and even roads do not exist. The health center is within a six hours walking distance and only sparsely equipped. In general, the community members consult a doctor only in severe cases. Complaints of diarrhea are judged as normal conditions. Most of the indigenous people subsist from agriculture, still using feces as fertilizer. Animals live close together with the population, having access to hygiene-related infrastructures of the houses like kitchens. Furthermore, climate conditions favor infectious diseases, too, as humidity and temperatures are high, supporting the multiplication of microbial pathogens [1]. Such conditions are also optimal for the developmental cycle of *Cyclospora cayetanensis,* a protozoan parasite from the phylum coccidia causing enteric human disease [2]. Transmission occurs via the fecal–oral route due to the consumption of contaminated food, particularly of fresh fruit or vegetables [2], or of contaminated drinking water [3], but not directly from human to human, because unsporulated oocysts need to sporulate in environmental compartments like water or soil first [2]. Density of water contamination with *C. cayetanensis* is correlated with disease prevalence in humans, while contact with poultry and contaminated soil as well as poor sanitation have been identified as risk factors for cyclosporiasis [4,5], athough there is no evidence of a non-human host for *Cyclospora cayetanensis*. 

Primary site of human infection is the upper small intestinal tract [6]. While *C. cayetanensis*-associated gastroenteritis is usually self-limiting after partly prolonged courses in immunocompetent individuals, long-lasting severe diarrhea may occur in immunocompromised patients, sometimes with relapses despite cotrimoxazole therapy [2,3,7,8]. Clinical courses about several weeks or months have been described without antimicrobial treatment. Thereby, recorded symptoms comprised watery diarrhea, anorexia, nausea, weight loss, and enteric malabsorption associated with villous atrophy and crypt hyperplasia [3,9]. In addition to active infection, various potential sequelae like Guillain-Barré syndrome, reactive arthritis and acalculous cholecystitis have been associated with *C. cayetanensis* infections [6]. Next to symptomatic infections, however, asymptomatic colonization has been reported from high endemicity settings as well [10,11].

Although *C. cayetanensis* shows a worldwide distribution [2] with an estimated global average prevalence of 3.55% [7], primary areas of endemicity comprise resource-poor countries with limited or poor hygiene standards [2]. In those regions, foreigners, children, elderly, and immunocompromised individuals are particularly frequently diagnosed with cyclosporiasis, while returning travelers are affected in non-endemic areas without relevant effects of sex or age [2,12,13,14]. In particular, individuals with low socio-economic status are at risk of getting infected with *C. cayetanensis* [15,16] as impressively indicated by a prevalence of 22.2% in slum dwellers in South Chennai, India [17]. However, imported and insufficiently cleaned raw fruit and vegetables imported from areas of endemicity are sources of infection even in non-endemic countries and irrespective of socio-economic status [2,8,9,18]. Accordingly, large foodborne cyclosporiasis outbreaks have been described in industrialized countries as well [19,20]. Food safety management systems are important elements in the prevention of cyclosporiasis [21,22,23]. A considerable hygiene-relevance of *C. cayetanensis* has been impressively demonstrated by outbreaks in travelers [24], on cruise ships [25], and by the demonstration of *C. cayetanensis* in tap water on trains even in an industrialized country [26].

Interestingly, there seem to be seasonal trends in the epidemiology of cyclosporiasis, although they have been inconsistently reported over different geographic regions [2,6]. As known from other reports, fecal–oral infections are particularly frequent in the moist hot weather of the rainy seasons in the tropics [27]. Cyclosporiasis is no exemption with peaks in the rainy season or in the summer [7].

Under the microscope, *C. cayetanensis* are non-refractile double-walled spheres with a diameter of 8–10 µm [3]. Although the oocysts are also visible on plain wet mounts, microscopical diagnosis can be facilitated by modified acid-fast stain with variable red staining of the cells after formalin fixation [3,28] and by concentration techniques [29]. Also, the use of autofluorescing properties of oocysts when exposed to ultraviolet illumination has been suggested to decrease the probability of overlooking them [30,31]. Despite such strategies, due to the lack of sensitivity of microscopical routine diagnosis, cyclosporiasis is believed to be underdiagnosed [32]. More recently, however, molecular diagnostic approaches have been associated with higher sensitivity, if available [7,33,34]. Even broad automated multiplex-polymerase chain reaction (PCR)-panels including *C. cayetanensis* can be applied [35].

Studies on the epidemiology of *C. cayetanensis* can best be performed in endemic areas, e.g., in Central and South American countries [1,36,37], where fecal–oral transmission via food sources [38] even beyond specific outbreak situations [39] is likely. In the assessment presented here, the effects of seasonality on the prevalence of *C. cayetanensis* infections or colonization was investigated. In detail, we compare the *C. cayetanensis* prevalence in stool samples of Colombian indigenous people in a high endemicity setting as observed in the rainy season [1] with the prevalence as assessed four years later in the dry season. By doing so, the period prevalence and the background colonization shall be assessed.

## 2. Materials and Methods

### 2.1. Study Type

The study was conducted as an epidemiological follow-up assessment in indigenous people living in a remote region of tropical Colombia, where a high baseline colonization rate with *C. cayetanensis* in the rainy season was known from a previous cross-sectional study [1]. Four years after this first assessment, a second stool sample collection was conducted in the dry season to assess the dimension of seasonal effects on the colonization rate with *C. cayetanensis*. The first collection had been performed between July and November (rainy season) 2014, the second was conducted between January and April 2018 (dry season).

### 2.2. Study Population

The first study population (2014) consisted of 137 stool samples from indigenous people living in the villages Tezhumake (81 samples), Department Cesar, and Siminke (43 samples), Department La Guajira. A further 13 samples came from a nearby village, Valledupar, Department Cesar.

The second study (2018) was conducted again in Tezhumake (168 samples) and Siminke (35 samples), but also in Cherua (91 samples), Department Cesar and in Ashintukwa (52 samples), Department La Guajira.

Recorded data comprised age, sex, height, weight, and self-reported gastroenteric symptoms (comprising abdominal pain and/or diarrhea) at the time of the sample collection. Stool collectors were provided after the examination and given back the same or the next day. The samples were brought to the laboratory directly after collection at each day of collection for further processing.

Direct comparisons of individuals at different time points were unfeasible due to the need for anonymization as demanded for ethical reasons.

### 2.3. Microscopy and Real-Time Polymerase Chain Reaction (PCR)-Based Screening for Cyclospora cayetanensis

One part of the each stool samples was used for microscopy, the other part was stored frozen at −20 °C prior to nucleic acid extraction using the QIAamp DNA stool mini kit (QIAGEN, Hilden, Germany) as recommended by the manufacturer. Extraction was performed a few weeks to months after each collection period. Real-time PCR for *C. cayetanensis* was performed according to the protocol by Verweij and colleagues [40] with the adaptation as detailed by Frickmann and colleagues [41] on RotorGene Q cyclers (Qiagen). Within each run, a negative control based on PCR-grade water and a positive control based on a plasmid as previously described [41] were included. As calculated with a dilution series of the positive control plasmid and the software SciencePrimer.com (http://scienceprimer.com/copy-number-calculator-for-realtime-pcr, accessed on 2 February 2021), a detection limit of 51 copies/µL eluate was defined. Inhibition control was based on a real-time PCR targeting Phocid Herpes Virus (PhV) DNA as described previously [42].

### 2.4. Statistics

Due to the low number of obtained samples, the assessment was restricted to descriptive statistical analyses. Significance was calculated applying simple but robust procedures like Fisher’s exact test for binary datasets and Mann–Whitney U-testing or Kruskal–Wallis testing for quantitative assessments. The software GraphPad Instat version 3.06 (GraphPad Software Inc., La Jolla, CA, USA) was used for the calculations.

### 2.5. Ethics

Ethical clearance for the initial assessment in 2014 was provided by the Ethics Committee of Valledupar, Cesar, Colombia (Acta no 0022013, provided in February 2013). For the follow-up assessment, ethical clearance was guaranteed by the Ethics Committee for Research in Santa Marta (Acta no 102016, provided in October 2016). Written informed consent was obtained from each participant or from the parent or legal guardian of children before participation. The study was performed in agreement with the principles of the Declaration of Helsinki.

## 3. Results

### PCR-Based Detection of Cyclospora cayetanensis and Associated Patient-Specific Features

As shown in Table 1, with 16/136 (11.8%), a considerably high proportion of tested individuals was positive by PCR for *C. cayetanensis* in the rainy season compared to only 15/292 (5.1%) in the dry season. While the *C. cayetanensis* cases were equally distributed among males and females, significance for low colonization rates in the dry season could be confirmed for the female sex only. The compared populations were homogeneously distributed regarding age but not regarding size and weight (Table 1). Relatively more patients with gastrointestinal symptoms and positive *C. cayetanensis* PCR results were recorded in the dry season, although the absolute numbers were low in both assessed populations. Lower Ct values indicate higher pathogen loads, which was found for the group tested during the rainy season. 

## 4. Discussion

The study was conducted to assess the effects of seasonality on the colonization or infection with *C. cayetanensis* in an indigenous population in the remote Colombian territories. While high endemicity during the rainy season has previously been described [1], a follow-up assessment about 4 years later in the dry season demonstrated a reduced but still relevant colonization rate.

We recorded infection rates with *C. caytanensis* varying between 5.4% (dry season) and 11.8% (rainy season) in the indigenous population. In comparison to the worldwide prevalence of 3.55% [7], these rates are way higher and demonstrate a need for hygienic countermeasures.

Higher measured Ct values in the dry season indicate lower parasite density. This has been suggested as a hint for lower pathogenicity or for the discrimination between infection and sole colonization [43,44,45]. However, contradicting results from other studies in high-endemicity settings make this assumption controversial [10,46,47]. 

In our study, only 2 out of 17 volunteers reported gastrointestinal symptoms in the rainy season despite higher pathogen density as indicated by lower Ct-values. In contrast, 6 out of 15 volunteers with *C. cayetanensis* detections reported such symptoms in the dry season despite higher Ct values associated with lower parasite load. These findings, however, have to be interpreted with care, because gastrointestinal disease can be caused by many different agents and pathogen combinations in areas with low hygiene standards as previously reported for Colombian indigenous people [1]. Further it has to be kept in mind that in the Wiwa communities, complaints on disorders like diarrhea or abdominal pain are only mentioned sparsely. In particular, diarrhea is not considered as a serious condition, because it is very common in the indigenous people. However, the low absolute number of 2 patients with proof of *C. cayetanensis* and gastrointestinal symptoms in the rainy season suggests a high proportion of subclinical colonization due to this pathogen in the population assessed.

The study has a number of limitations. First, while the groups with and without *C. cayetanensis* detection were well balanced regarding the parameter age, differences regarding size, weight and the proportion of *C. cayetanensis* positivity in females during both study intervals might represent a source of bias. Second, the numbers of the assessed individuals during the two assessments were not identical and still quite low, which makes the interpretation challenging. Third, the applied PCR assays show imperfect performance characteristics regarding both sensitivity and specificity as recently described [43]. However, as identical assays were applied for both assessments, the results should nevertheless remain comparable, although minor changes in the composition of the nucleic acid extraction kits might have shown interfering effects. Fourth, the performed discrimination between colonization and infection was just an estimation, because a clear cut-off was impossible to define under the limited investigational conditions in the remote villages. Fifth, an interval of about 3 years between both assessments makes it theoretically possible that factors different from seasonality might have affected the recorded prevalence values. However, as the living conditions of the indigenous did not relevantly change in the meantime, potential effects of improved hygiene procedures are at least not very likely. Sixth, the low numbers of inhabitants of the villages made stratification for homogeneity of factors other than comparable living conditions unfeasible.

## 5. Conclusions

In spite of the aforementioned limitations, the study shows, that the high endemicity of *C. cayetanensis* in Colombian indigenous people in the rainy season still remains considerable also in the dry season despite reduced parasite loads and lower case-numbers. Lacking association between Ct values and proportions of individuals with *C. cayetanensis* and gastrointestinal symptoms suggests predominantly enteric colonization in this population rather than enteric infection. The striking difference between the proportions of individuals with gastrointestinal symptoms during the rainy season with and without PCR detections of *C. cayetanensis* makes an etiological role of other infectious agents, which are highly abundant in the assessed populations as shown previously [1], more likely.

## Figures and Tables

**Table 1 microorganisms-09-00627-t001:** Polymerase chain reaction (PCR)-based detection of *Cyclospora cayetanensis* in the course of the two screenings in 2014 and 2018 as well as associated patient data.

	Screening in the Rainy Season of 2014 (n = 137 Individuals Screened for *C. cayetanensis*)		Screening in the Dry season of 2018 (n = 292 Individuals Screened for *C. cayetanensis*)		Significance *
	Positive for *C. cayetanensis*	Negative for *C. cayetanensis*	Positive for *C. cayetanensis*	Negative for *C. cayetanensis*	(threshold 0.05)
Numbers (n) of individuals	17	120	15	277	0.0102 *
Number of males	7	60	8	122	0.3949
Number of females	10	60	7	155	0.0236 *
Age (mean ± standard deviation SD)	23.2 (±25.7)	25.6 (±17.1)	30.1 (±22.2)	22.8 (±17.8)	0.1661
Size (in cm, mean ± standard deviation SD)	121.6 (±30.7)	138.8 (±22.4)	127.8 (±29.3)	127.0 (±36.1)	0.0008 *
Weight (in kg, mean ± standard deviation SD)	32.3 (±17.3)	43.6 (±15.5)	39.8 (±24.3)	32.8 (±20.2)	0.0014 *
Recorded gastro-intestinal symptoms (n/n, %)	2/17 (11.8%)	46/120 (38.3%)	6/15 (40.0%)	16/277 (5.8%)	<0.0001 *
Cycle threshold values in real-time PCR (mean ± standard deviation SD)	30.3 (±3.4)	n.a.	34.2 (±1.4)	n.a.	0.0008 *

* = Significant differences, which were calculated applying Fisher’s exact test for binary datasets and Mann–Whitney U-testing (Ct values) or Kruskal–Wallis testing (age, weight, size) for quantitative assessments., n.a. = not applicable.

## Data Availability

All relevant data are provided in the manuscript.

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
