# Peer review of "Seasonal Differences in Cyclospora cayetanensis Prevalence in Colombian Indigenous People"

_microorganisms, 2021, doi:10.3390/microorganisms9030627_

Round 1

Reviewer 1 Report

This is a well written paper.

line 53-replace cysts with oocysts

line 93-replace oozysts with oocysts

Author Response

This is a well written paper.

line 53-replace cysts with oocysts

line 93-replace oozysts with oocysts

Thank you for these hints, we corrected the words in both lines.

Reviewer 2 Report

Title: I believe it is more appropriate to say "indigenous people", please correct throughout the manuscript

20- suggest replacing "enteric parasitosis caused by Cyclospora cayetanensis" with cyclosporiasis

22- I don't think that you can distinguish between colonization and infection based on the data that you collected

29- replace less with "fewer"

39- water is "obtained"

41-health "care centre" rather than "health point"

43-suggest changing to "complaints of diarrhea" 

44-suggest "make a living from" or "subsist from" agriculture, and replace "top dressing" (vague) with manure/faeces or other more familiar term if that is what is intended. 

52- "oocysts" not cysts

55- There is no evidence of a non-human host for Cyclospora cayetanensis, this should be clearly stated. 

93-"oocysts"

101- use "like" or "e.g" not both

106-Was it 3 or 4 years? This is a very long period between collections to be directly comparing the impact of wet vs dry season. There are many other factors that could be contributing to the differences between these collections-water sources, animal/farming practices, change in population, etc.

125-what GI symptoms were recorded/collected? Diarrhea? Abdominal pain? Were any of the individuals the same from either collection period? Could individual results be compared? 

129-There were changes to the QIAamp Stool kit during the time between this study's two collections, were all of the samples tested at the same time or were there several years in between? How long were they stored prior to testing? 

159-How did you determine (statistically) that the populations were homogeneous?

From reference #1: "The 137 questionnaires and physical examinations revealed that 88% (119) had no complaints, 10% (13) complained about diarrhea, 2% (2) noted abdominal pain, and 2% (2) suffered from diarrhea and abdominal pain. In one case, information was missing." These numbers do not agree with what is presented here, where you indicated that 48/137 reported GI symptoms. Can you explain the discrepancy? 

Table 1-which means differ? Please mark with subscript letters/numbers.

176-177-commas instead of decimals

Many protocols for extraction of DNA from oocysts use multiple freeze thaw cycles or bead-beating to disrupt the oocyst wall. This was not perofrmed here prior to DNA extraction-could this have impacted the rate of detection? 

192-If GI symptoms are poorly reported, then how reliable is your conclusion that "subclinical colonization" is more common in the rainy season vs dry season? Can you discuss other reasons that more people reported GI symptoms in the rainy vs dry season (ie in reference #1 you reported prevalence of other parasites/bacteria in the same set of samples)?

Author Response

Title: I believe it is more appropriate to say "indigenous people", please correct throughout the manuscript

Thank you for this hint, we changed it in the title and throughout the paper

20- suggest replacing "enteric parasitosis caused by Cyclospora cayetanensis" with cyclosporiasis

This was done.

22- I don't think that you can distinguish between colonization and infection based on the data that you collected

We agree, a clear-cut discrimination is impossible. Based on the differences in reported symptoms, we have just estimated that colonization might be more likely than infection. Of course, such estimations may apply on population level but hardly for the individual. A respective comment has been added as a new fourth limitation at the end of the discussion.

29- replace less with "fewer"

39- water is "obtained"

41-health "care centre" rather than "health point"

This was done. In line with American English style, “center” rather than “centre” was written.

43-suggest changing to "complaints of diarrhea" 

44-suggest "make a living from" or "subsist from" agriculture, and replace "top dressing" (vague) with manure/faeces or other more familiar term if that is what is intended. 

Changes in line 29, 39, 41, 43, 44 were performed. Thank you for the good advices.

52- "oocysts" not cysts

This was corrected.

55- There is no evidence of a non-human host for Cyclospora cayetanensis, this should be clearly stated. 

We agree and have added this point.

93-"oocysts"

“Parasites” has been replaced by “oocysts” as requested and we have corrected the typing error.

101- use "like" or "e.g" not both

The abbreviation e.g. was deleted.

106-Was it 3 or 4 years? This is a very long period between collections to be directly comparing the impact of wet vs dry season. There are many other factors that could be contributing to the differences between these collections-water sources, animal/farming practices, change in population, etc.

As the collections were performed over several months each time, the difference was between 3 and 4 years (between 2014 and 2018 as stated under the methods sub-heading “2.2. study population”). The time span has been necessary for reasons of funding restraints. We agree that the mentioned factors may have influenced the findings. However, as the living conditions of the people have not changed in the meantime, we feel that such effects should not be overestimated. Nevertheless, we have added a respective 5thlimitation at the end of the discussion.

125-what GI symptoms were recorded/collected? Diarrhea? Abdominal pain? Were any of the individuals the same from either collection period? Could individual results be compared? 

We have specified that indeed abdominal pain and/or diarrhea were meant as we have now stated under the methods sub-heading “2.2. study population”. As a new last sentence of this sub-heading, we have added that: “ Direct comparisons of individuals at different time points were unfeasible due to the need of anonymization as demanded for ethical reasons”.

129-There were changes to the QIAamp Stool kit during the time between this study's two collections, were all of the samples tested at the same time or were there several years in between? How long were they stored prior to testing? 

We have now stated under the methods sub-heading “2.3. Real-time PCR-based screening for C. cayetanensis” that.”Extraction has been performed a few weeks to months after each collection period”. We confirm that there have been minor modifications to the QIAamp Stool kit in the meantime, but to our experience, its reliability hasn’t relevantly changed due to those modifications. Nevertheless, we agree, that this is a potential weakness, so we have added this point to the third limitation at the end of the discussion.

159-How did you determine (statistically) that the populations were homogeneous?

Due to the low numbers of inhabitants in the villages, stratification for homogeneity of factors other than comparable living conditions was unfeasible. We have added this sixth limitation at the end of the discussion.

From reference #1: "The 137 questionnaires and physical examinations revealed that 88% (119) had no complaints, 10% (13) complained about diarrhea, 2% (2) noted abdominal pain, and 2% (2) suffered from diarrhea and abdominal pain. In one case, information was missing." These numbers do not agree with what is presented here, where you indicated that 48/137 reported GI symptoms. Can you explain the discrepancy? 

The reason is, that the intensity of physical examination differed between the assessments of the two populations. This avoid reporting bias, only self-reported gastrointestinal symptoms were included in the present assessment while the results of physical examinations were not considered. We make this point more clear by stating that only “self-reported” symptoms were assessed under the methods sub-heading “2.2. study population”.

Table 1-which means differ? Please mark with subscript letters/numbers.

As already stated in the last column, significance threshold was 0.05. We marked this now with an asterix, which enlightens the significant differences. This means that difference was observed for all parameters with the exception of “numbers of males” and “age”.

176-177-commas instead of decimals

The adjusted this.

Many protocols for extraction of DNA from oocysts use multiple freeze thaw cycles or bead-beating to disrupt the oocyst wall. This was not performed here prior to DNA extraction-could this have impacted the rate of detection? 

We agree that slightly higher sensitivity might have been achieved if other than standard nucleic acid extraction procedures had been applied. As, however, this sensitivity limitation affects both assessed sample populations in the same way, the recorded difference between the compared populations should still be valid.

192-If GI symptoms are poorly reported, then how reliable is your conclusion that "subclinical colonization" is more common in the rainy season vs dry season? Can you discuss other reasons that more people reported GI symptoms in the rainy vs dry season (ie in reference #1 you reported prevalence of other parasites/bacteria in the same set of samples)?

In spite of the overall poor reporting, there is still considerably higher reporting of gastrointestinal symptoms during rainy season than during dry season. In spite of this higher overall percentage of gastrointestinal symptoms during rainy season, percentage of individuals with C. cayetanensis detections in PCR report much lower proportions of gastrointestinal symptoms, even compared with the dry season. So, we consider it justified at least to assume that C. cayetanensis might have predominantly played as role as colonizers, while the reported gastrointestinal symptoms were more likely attributed to other gastrointestinal pathogens. As requested, we add a new last sentence to the conclusions, stating: The striking difference between the proportions of individuals with gastrointestinal symptoms during the rainy season with and without PCR detections of C. cayetanensis makes an etiological role of other infectious agents, which are highly abundant in the assessed populations as shown previously [1], more likely”.

Round 2

Reviewer 2 Report

All comments from first review have been addressed. I would suggest including some additional details on how the samples were collected. 

Author Response

All comments from first review have been addressed. I would suggest including some additional details on how the samples were collected. 

Thank you very much for this hint. We added in the section 2.2 and 2.3 the following sentences: „Stool collectors were provided after the examination and given back the same or the next day. The samples were brought to the laboratory directly after collection at each day of collection for further processing“.

„One part of each stool sample was used for microscopy, the other part was stored frozen at -20°C prior to nucleic acid extraction using the QIAamp DNA stool mini kit (QIAGEN, Hilden, Germany) as recommended by the manufacturer“.